# A tree based eXtreme Gradient Boosting (XGBoost) machine learning model to forecast the annual rice production in Bangladesh

Mst Noorunnahar[1⦾], Arman Hossain Chowdhury[2⦾], Farhana Arefeen Mila[3]*

1 Department of Statistics, Bangabandhu Sheikh Mujibur Rahman Agricultural University, Gazipur, Bangladesh, 2 Department of Statistics, Begum Rokeya University, Rangpur, Bangladesh, 3 Department of Agribusiness, Bangabandhu Sheikh Mujibur Rahman Agricultural University, Gazipur, Bangladesh

⦾ These authors contributed equally to this work.
* famila@bsmrau.edu.bd

**Data Availability Statement:** All necessary codes and data are available on GitHub (https://github.com/Arman-Hossain-Chowdhury/Rice-production).

## Abstract

In this study, we attempt to anticipate annual rice production in Bangladesh (1961–2020) using both the Autoregressive Integrated Moving Average (ARIMA) and the eXtreme Gradient Boosting (XGBoost) methods and compare their respective performances. On the basis of the lowest Corrected Akaike Information Criteria (AICc) values, a significant ARIMA (0, 1, 1) model with drift was chosen based on the findings. The drift parameter value shows that the production of rice positively trends upward. Thus, the ARIMA (0, 1, 1) model with drift was found to be significant. On the other hand, the XGBoost model for time series data was developed by changing the tunning parameters frequently with the greatest result. The four prominent error measures, such as mean absolute error (MAE), mean percentage error (MPE), root mean square error (RMSE), and mean absolute percentage error (MAPE), were used to assess the predictive performance of each model. We found that the error measures of the XGBoost model in the test set were comparatively lower than those of the ARIMA model. Comparatively, the MAPE value of the test set of the XGBoost model (5.38%) was lower than that of the ARIMA model (7.23%), indicating that XGBoost performs better than ARIMA at predicting the annual rice production in Bangladesh. Hence, the XGBoost model performs better than the ARIMA model in predicting the annual rice production in Bangladesh. Therefore, based on the better performance, the study forecasted the annual rice production for the next 10 years using the XGBoost model. According to our predictions, the annual rice production in Bangladesh will vary from 57,850,318 tons in 2021 to 82,256,944 tons in 2030. The forecast indicated that the amount of rice produced annually in Bangladesh will increase in the years to come.

## Introduction

There has been a fast expansion in the world population, which has put a strain on the agricultural sector [1]. Rice is considered the world's third most common major crop, with more than

**Funding:** The author(s) received no specific funding for this study.

**Competing interests:** The authors have declared that no competing interests exist.

50% of the world's population eating it as a staple diet [2, 3]. As one of the most nutrient-dense grains, rice is an excellent source of carbohydrate as well as vitamins (B, E, thiamine) and minerals (Ca, Mg, Fe) [4]. About 160 million Bangladeshis rely on rice as a basic meal for their daily diets and survival [5]. Bangladesh's economy is heavily dependent on rice production, which means that the price of rice has a considerable impact on GDP growth, inflation, wages, employment, food security, and poverty [6]. The rice industry employs over 48% of the rural population, provides two-thirds of all caloric intake, and accounts for half of the average person's protein intake [7]. For agricultural GDP and national income, the rice subsector alone contributes about 4.5% to the GDP [8]. Nearly all farming households in Bangladesh cultivate rice. It is produced on about 10.5 million hectares of land, which occupies about 75 and 80% of the total cropped and irrigated areas, respectively [9].

Accurate and timely estimates of crop production before harvest are essential for food security and administrative planning, especially in the current, ever-changing global environment and international scenario [10, 11]. Rice yield forecasting has been extensively examined using various methods all around the world. In order to forecast rice yield, Kumar and Kumar (2012) added fuzzy values to the time series [12]. Alam et al. (2018) applied two hybrid approaches including ARIMAX-ANN and ARIMAX-SVM for estimating rice yield in India [13]. Jing-feng (2011) used NOAA/AVHRR data to predict rice production in Zhejiang Province through ratio models and regression models [14]. Using a crop growth model, Yun (2003) forecasted regional rice production in South Korea [15]. Koide et al. (2013) employed precipitation hindcasts from one uncoupled general circulation model (GCM) and two coupled GCMs to examine the predictive abilities of retrospective seasonal climate forecasts (hindcasts) customized to Philippine rice production data [16]. A satellite remote sensing technique was used by Noureldin et al. (2013) to forecast the production of rice in Egypt [17]. However, to reveal the growth pattern and make the most accurate prediction of rice production in Bangladesh, it is necessary to use a suitable approach that can successfully describe the observed data. Different techniques have been taken to accurately estimate yield, and each method has its own strengths and limitations [18]. For example, Rahman (2010), Mahmud (2018), Rahman et al. (2016), and Sulatana and Khanam 2020 applied the autoregressive integrated moving average (ARIMA) and artificial neural network (ANN) for predicting rice production in Bangladesh [19–22].

Sensor technologies, big data, the Internet of Things, artificial intelligence (AI), and machine learning approaches have recently shown great potential to advance precision agriculture and obtain accurate predictions [23]. According to the aforementioned literature and to the best of the author's knowledge, XGBoost is a machine learning algorithm that has not been widely deployed. The eXtreme Gradient Boosting (XGBoost) model is a supervised machine learning technique and an emerging machine learning method for time series forecasting in recent years [24, 25]. It is a novel gradient tree-boosting algorithm that offers efficient out-of-core learning and sparsity awareness. XGBoost is a supervised learning technique that ought to be particularly good for the problem of claim prediction with both big training data and missing values, even if the commonly used methods such as random forest and neural networks can handle missing values [26, 27]. The robustness of XGBoost results in increased usage of the method in many other applications. As an example, Aler et al. utilize XGBoost in the field of direct-diffuse solar radiation separation by creating two models [28]. Moreover, in infectious disease prediction such as COVID-19, the XGBoost achieved greater prediction accuracy [29, 30].

In contrast, the Autoregressive Integrated Moving Average (ARIMA) model developed by Box and Jenkins (1990) is most widely used for forecasting time series data because of its capacity to handle non-stationary data [31]. The ARIMA model is a suitable forecasting

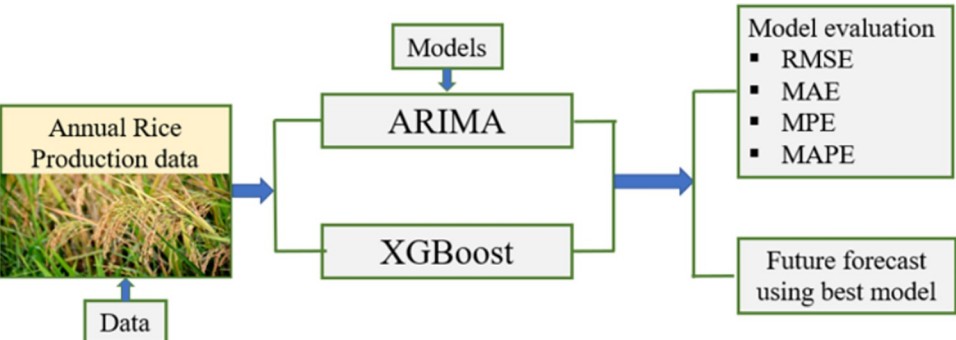

**Fig 1. Theoretical framework for the study.**

method in agriculture for different crops and has been extensively used in the fields of economics and finance [31–33]. Therefore, this study aimed to (a) compare the predictive accuracy of the autoregressive integrated moving average (ARIMA) and eXtreme gradient boosting (XGBoost) for accurate modeling the annual rice production data in Bangladesh; and (b) carry out the best model to forecast rice production for the next 10 years (Fig 1). Finally, the findings of this study will help government officials and development practitioners make more accurate short-term predictions of future rice production to boost administrative planning and ensure food security.

## Materials and methods

### Data source

The annual rice production data from 1961 to 2020 (60 years) used in this study were collected from the website of FAOSTAT [34]. The data were divided into training and test sets. The proportion of training and testing data was 90% and 10%, respectively. The ARIMA and XGBoost models were built using the training data sets. The test data were used to evaluate the predictive ability of the developed models. The data set does not contain any missing values.

### ARIMA model

The autoregressive integrated moving average (ARIMA) is a technique for analyzing and predicting time series data that was initially introduced by Box and Jenkins in 1976 [35]. An ARIMA (p, d, q) time series model consists of its three components. The letters p of the ARIMA model denote the autoregressive (AR) order, d denotes the differencing order, and q denotes the moving average order (MA) [36, 37]. The autoregressive order AR(p) describes the linear combination of the observations that are p times earlier with the random shock term, which can be mathematically defined as

$$Y_t = C + \emptyset_1 Y_{t-1} + \emptyset_2 Y_{t-2} + \emptyset_3 Y_{t-3} + \emptyset_4 Y_{t-4} \ldots \ldots \emptyset_p Y_{t-p} + \varepsilon_t \tag{1}$$

Where, $Y_t$ and $\varepsilon_t$ represent the observed value and the random shock terms at time t, $\emptyset_i$ (i = 1,2,3,4. . ..) indicates the model parameters, and c is the constant term. On the other hand, the moving average order MA(q) explains the dependent variable for previous random shock terms, which can be defined as

$$Y_t = \mu + \theta_1 \varepsilon_{t-1} + \theta_2 \varepsilon_{t-2} + \theta_3 \varepsilon_{t-3} + \theta_4 \varepsilon_{t-4} + \cdots + \theta_q \varepsilon_{t-q} + \varepsilon_t \tag{2}$$

where, $\mu$ represents the mean of the series, $\theta_j$ (j = 1, 2, 3. . . q) denotes the model parameters,

and q indicates the model's order [38]. According to the above explanation, the ARMA (p, q) model can be defined mathematically as follows:

$$Y_t = C + \mu + \emptyset_1 Y_{t-1} + \emptyset_2 Y_{t-2} + \emptyset_3 Y_{t-3} + \emptyset_4 Y_{t-4} \ldots\ldots + \emptyset_p Y_{t-p} + \theta_1 \varepsilon_{t-1} + \theta_2 \varepsilon_{t-2} + \theta_3 \varepsilon_{t-3}$$
$$+ \theta_4 \varepsilon_{t-4} + \cdots + \theta_q \varepsilon_{t-q} + \varepsilon_t \tag{3}$$

The general form of the ARIMA (p, d, q) model with the differenced series may be defined mathematically as follows:

$$y'_t = c + \emptyset_1 y'_{t-1} + \emptyset_2 y'_{t-2} + \ldots + \emptyset_p y'_{t-p} + \theta_1 \varepsilon_{t-1} + \theta_2 \varepsilon_{t-2} + \ldots + \theta_q \varepsilon_{t-q} + \varepsilon_t \tag{4}$$

Where $y'_t$ explains the difference between the series (the number of differences can be greater than 1);; $\emptyset_1, \emptyset_2 \ldots \emptyset_p$ indicate the coefficients of AR(p) terms and $\theta_1, \theta_2 \ldots \theta_q$ show the coefficients of the moving average, MA(q) term. More information regarding ARIMA model can be found in the literature [30, 39].

## XGBoost model

The eXtreme Gradient Boosting (XGBoost) is a type of boosting application that combines several learning applications to produce higher prediction accuracy than any of the individual learning applications used in several fields [24]. It is a decision tree-based ensemble machine learning approach that is frequently employed in data science. After utilizing an internal approach that aggregates the outcomes from several individual trees, precise forecasts can be obtained [29]. XGBoost was first introduced by Chen Tianqi and Carlos in 2011, and since then several researchers have refined and enhanced it for the follow-up study [40]. The XGBoost model aims to execute a gradient descent optimization approach so that the loss function can be reduced [41]. Boosting is an ensemble technique that can assemble thousands of forecasting models with lower performance into a strong, high-performance model by repeatedly merging the models within permissible parameter values [40, 42]. The objective function can be written as follows:

$$obj(\theta) = \sum_i L(\hat{y}_i, y_i) + \sum_k \Omega(f_k) \tag{5}$$

As mentioned above, the objective function (5) consists of a loss function denoted by L and a regularization term $\Omega(f_k)$, that reduces the new tree's output variation. $\hat{y}_i$ denotes the predicted value and $y_i$ represents the observed value. A detailed information regarding the XGBoost model can be found in the literature [24, 39].

## Evaluation parameter of models

One of the major criteria of model evaluation is the calculation of model accuracy. The accuracy of a model describes how the actual and predicted values are close to each other. Model accuracy can be calculated by using several measures [43]. This study used the four widely used model accuracy measures, such as mean absolute percentage error (MAPE), mean percentage error (MPE), mean absolute error (MAE), and root mean square error (RMSE). These measures can be defined mathematically as follows:

$$MAE = \frac{1}{n} \sum_{i=1}^n |\hat{y}_i - y_i| \tag{6}$$

$$MPE = \frac{1}{n} \sum_{i=1}^n \left( \frac{\hat{y}_i - y_i}{y_i} \right) \times 100\% \tag{7}$$

$$RMSE = \sqrt{\frac{1}{n}\sum\nolimits_{i=1}^{n} (\hat{y}_i - y_i)^2} \tag{8}$$

$$MAPE = \frac{1}{n}\sum\nolimits_{i=1}^{n} |\frac{\hat{y}_i - y_i}{y_i}| \times 100\% \tag{9}$$

Where n indicates the number of samples, $\hat{y}_i$ denotes the predicted value and $y_i$ represents the observed value, and $\hat{y}_i - y_i$ indicates the error value. The MAPE measurement provides the percentage result of the errors. Better fitting results are achieved with less errors [41].

## Statistical analyses

ARIMA and XGBoost predictive models and several statistical analyses were carried out using RStudio (Version 4.2.1) [44]. The ARIMA model was fitted using the "forecast" package [45]. The XGBoost model was constructed with the "forecastxgb" package. The "ggplot2" package was used for graphical visualization. All necessary codes and data are available at https://github.com/Arman-Hossain-Chowdhury/Rice-production.

## Results

The highest amount of rice produced in Bangladesh was 54,905,891 tons in 2020, and the lowest was 13,304,520 tons in 1962. The average amount of rice produced annually in Bangladesh is 29,960,847.08 tons. And the boxplot indicates that the data have no outliers (Fig 2).

We plotted the time series of the annual rice production data from 1961 to 2020 in Bangladesh. The data vary considerably and show a linear pattern. The Augmented Dickey Fuller (ADF) test confirmed that the data are not smooth (Fig 3).

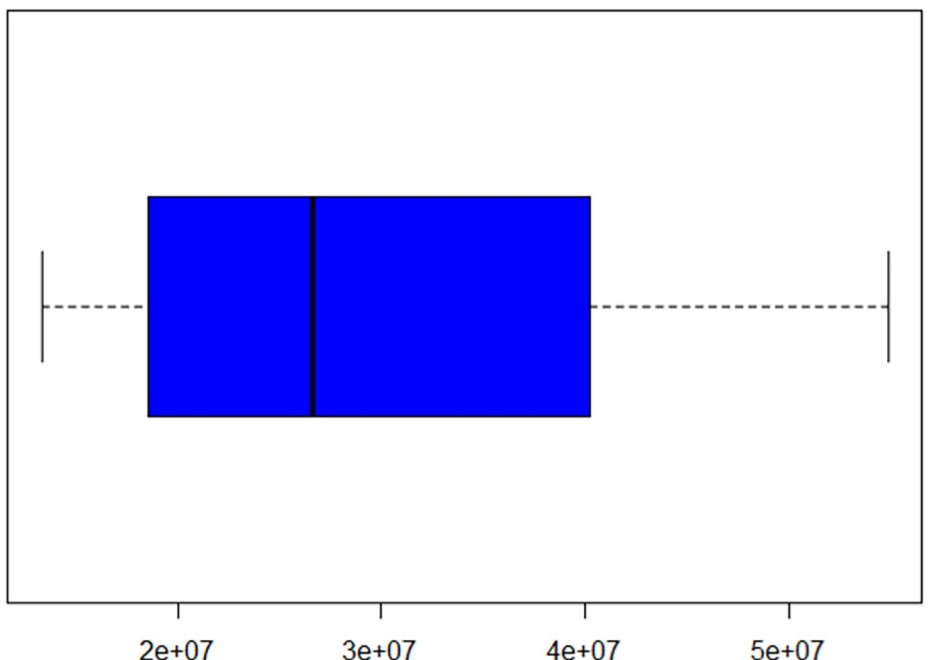

**Fig 2. Boxplot of the annual rice production data in Bangladesh from 1961 to 2020.**

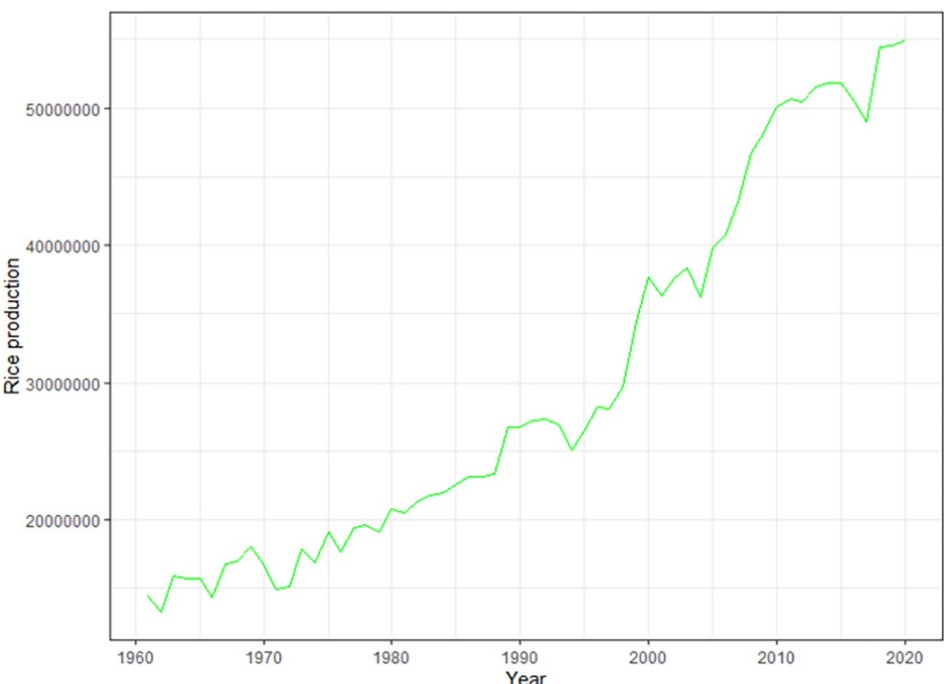

**Fig 3. A time series plot for rice production in Bangladesh from 1961 to 2020.**

To reduce variation and stabilize the actual data, Box & Cox (1964) presented a parametric power transformation technique [46]. We applied this technique to make the data stable and exhibit less variation (Fig 4) [47].

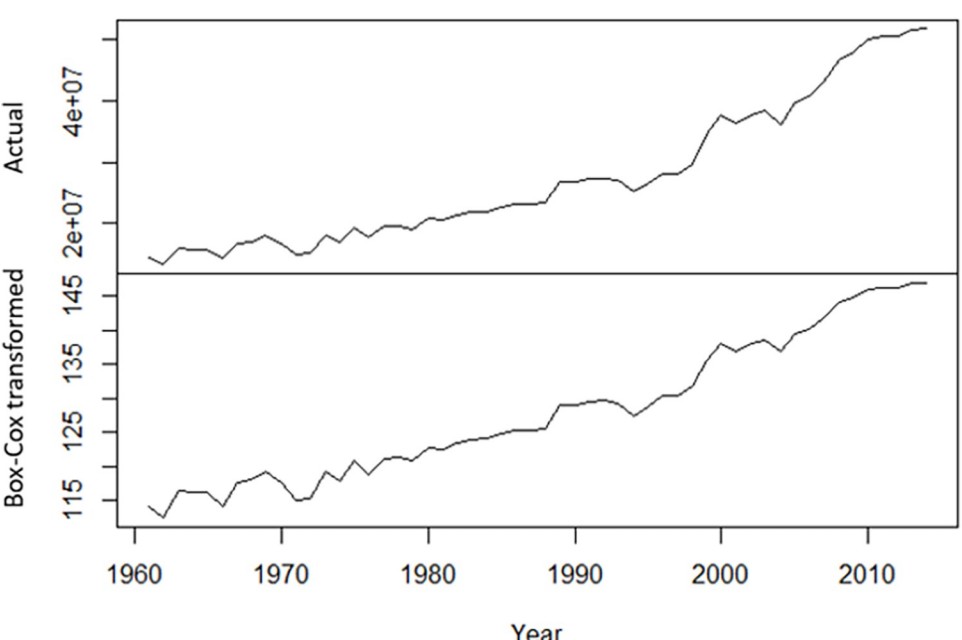

**Fig 4. A comparison between the Box-Cox transformed sequence and the original sequence of annual rice production in Bangladesh.**

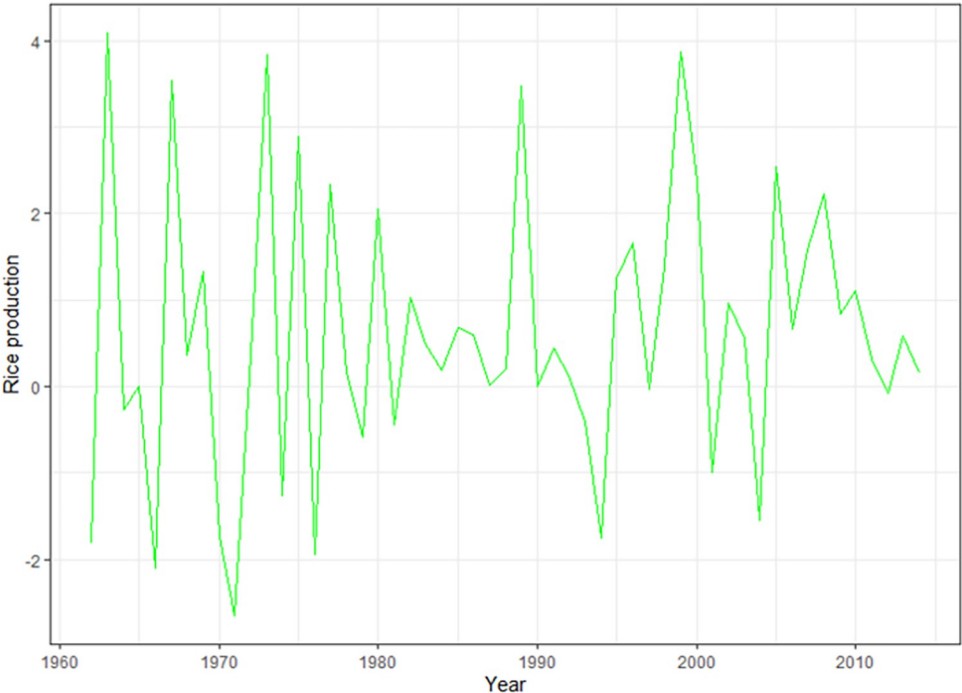

**Fig 5. First-order differencing of the rice production of the training data set shows stationarity.**

We performed the ADF test to see the stationarity of the data and found the data non-stationary (p-value = 0.57) at level. To compensate for the trend shift observed in (Fig 4), we used first-order differencing of the transformed sequence (Fig 5). The differenced time series was found stationary using the ADF test (p-value = 0.01). So, the parameter (d) of the ARIMA model was 1.

In the ACF diagram, there was an evident peak at lag 1 indicating that the MA may become 1 and an evident spike at lags 0 in the PACF diagram, suggesting that the AR may become 0 (Fig 6). Therefore, the maximum p and q values are 0 and 1, respectively.

The ARIMA model was built with the "auto.arima" function to list all possible models and then selected the model ARIMA (0,1,1) with drift on the basis of the lowest Corrected Akaikes Information Criteria (AICc) value. The drift parameter value indicates that the rice production drifts upward positively (Table 1).

After that, the residual diagram, the ACF diagram of the residual, and the residual histogram were drawn, indicating a normal distribution (Fig 7). Hence, the ARIMA (0, 1, 1) with drift model proved significant.

The XGBoost model was developed after adjusting several parameters. The adjusted parameters for the model were shown in S4 Table in S1 File. If a feature significantly affects the predicting performance when random noise takes its place, it is considered to be important. The feature importance of the XGBoost model was computed to see how each feature contributed to the prediction accuracy in the training set. And it was found that lag 5 of the training data contribute greatly to the model (Fig 8).

The curve of actual, fitted, and forecast values of the annual rice production in Bangladesh by ARIMA (0,1,1) with drift and the XGBoost model has been illustrated in Fig 9. The forecasted values of the XGBoost model were quite close to the actual values.

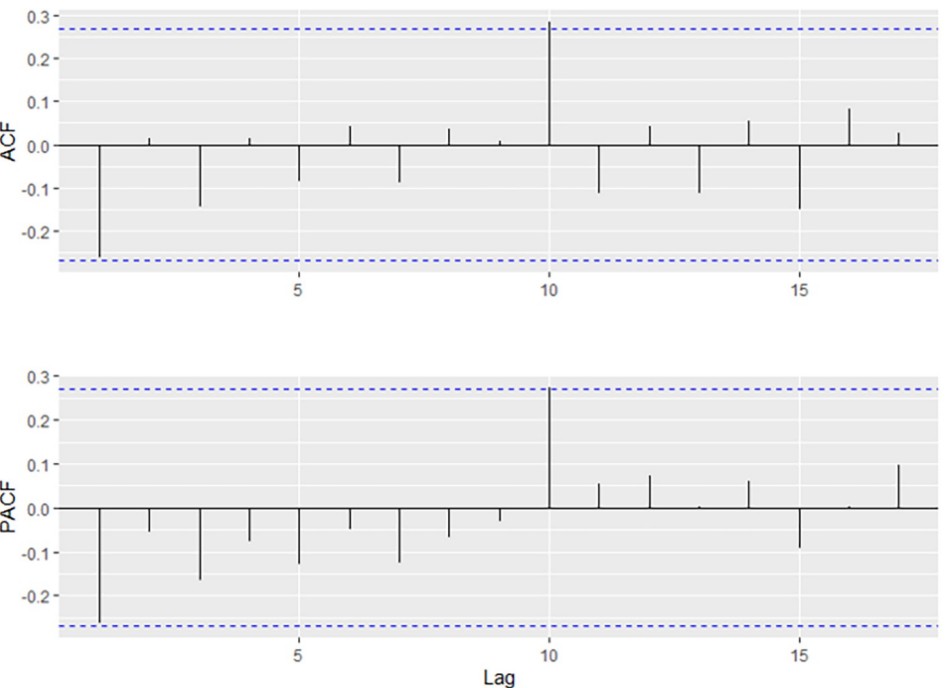

**Fig 6. The ACF and PACF diagram of rice production in Bangladesh after first order differencing.** ACF, autocorrelation function; PACF, partial autocorrelation function.

## Model comparison

The ARIMA (0,1,1) with drift model was built using the difference of the time series data. As a result, we lost a value in the training set; therefore, we compared the remaining 53 values. We used a maximum of eight time-lagged variables as input features for XGBoost. Because the maximum lag of 8 of the rice production data can contribute precisely to improve the XGBoost model prediction accuracy. Hence, the remaining 46 values were compared for the XGBoost model. The prediction accuracy for the ARIMA and XGBoost models is shown in Table 2.

The MAPE value of the test set of the XGBoost model was comparatively lower than the ARIMA model, which indicates that XGBoost performs better than ARIMA in predicting the annual rice production in Bangladesh. The detailed information regarding XGBoost model fitting can be found in S1 File.

Finally, based on our preferred XGBoost model, we predicted the annual rice production for the next 10 years (S1 File). According to our forecasts, during the next 10 years, the amount

**Table 1. Estimated parameters of the ARIMA (0,1,1) with drift model.**

| Parameters | Estimate | Std. Error | z value | Pr($>$|z|) |
|---|---|---|---|---|
| ma1 | -0.32448 | 0.15445 | -2.1008 | 0.03566* |
| drift | 0.62942 | 0.14259 | 4.4142 | 0.00001*** |
| AICc | 201.54 | | | |

AICc: Corrected Akaikes Information Criteria

Std. Error: Standard Error

ARIMA: Autoregressive Integrated Moving Average

Asterisk (*) indicates significant at 1% and (***) indicates significant at 0% level.

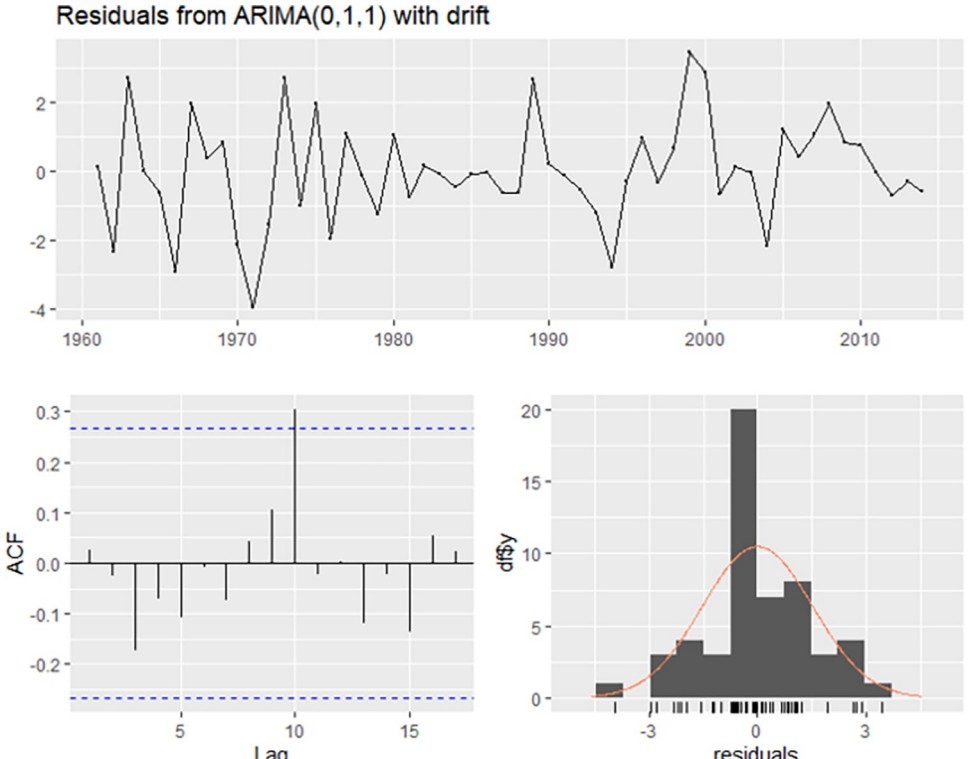

**Fig 7. A time series plot of the residuals with corresponding ACF diagram, and a histogram for the ARIMA (0,1,1) model with drift.** ACF, autocorrelation function; ARIMA, autoregressive integrated moving average.

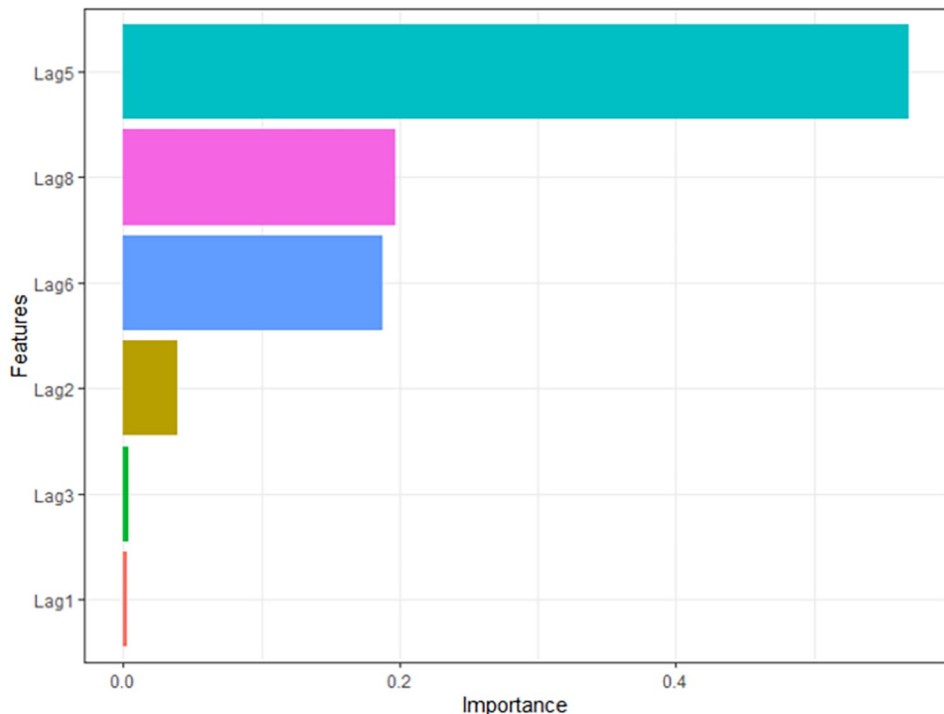

**Fig 8. Important characteristic features of the XGBoost model.**

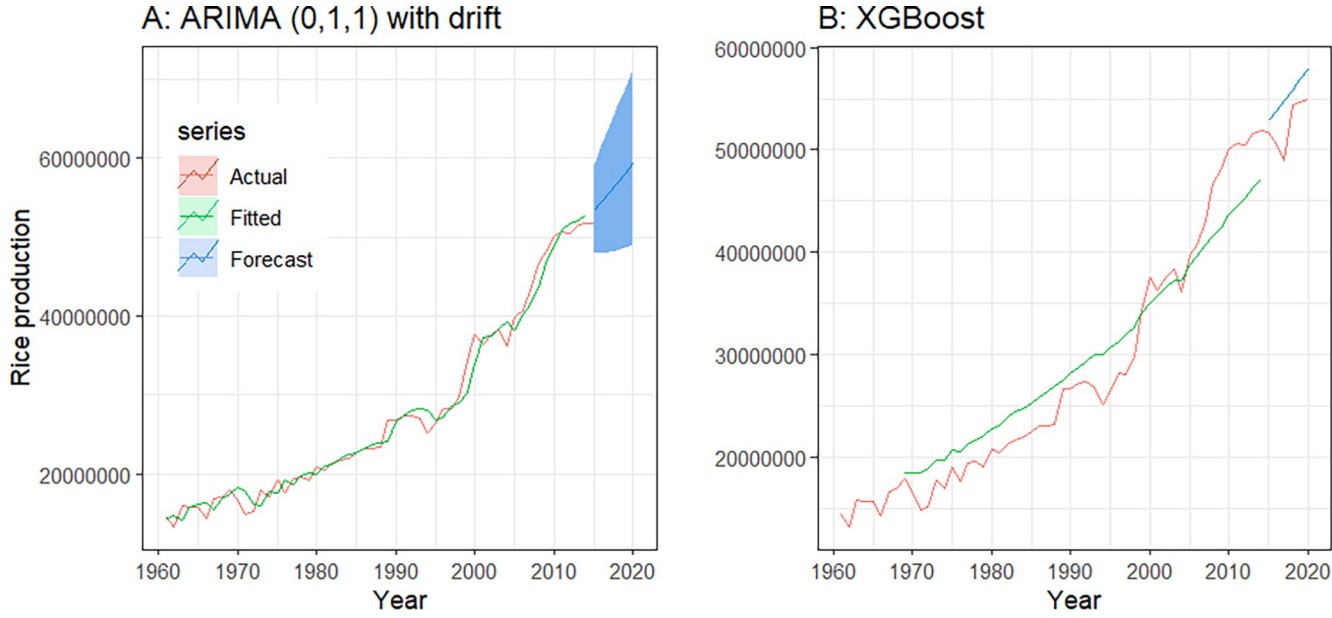

**Fig 9. ARIMA and XGBoost model show the actual, fitted and forecasted data for rice production in Bangladesh.** ARIMA, autoregressive integrated moving average; XGBoost, eXtreme Gradient Boosting.

of rice produced annually in Bangladesh will vary between 57,850,318 and 82,256,940 tons, as illustrated in Fig 10.

## Discussion

In our study, we found a linear upward pattern in the annual rice production data in Bangladesh. The primary goal of this study was to compare and contrast the predictive accuracy of the ARIMA and XGBoost forecasting models and make a short-term prediction with the best model. In this research, we examined the annual rice production in Bangladesh as a whole from 1961 to 2020. It is commonly known that Bangladesh has a subtropical tropical monsoon, which is distinguished by significant seasonal changes in precipitation, high temperatures, and humidity. In Bangladesh, there are three different seasons: a warm, humid summer from March to June; a chilly, wet monsoon season from June to October; and a cool, dry winter from October to March. In the past, temperatures in Bangladesh have ranged from 15°C to

**Table 2. Evaluation of parameters for the ARIMA and XGBoost model for rice production in Bangladesh.**

| Models | Training set | | | | Test set | | | |
|---|---|---|---|---|---|---|---|---|
| | **MAE** | **MPE** | **RMSE** | **MAPE** | **MAE** | **MPE** | **RMSE** | **MAPE** |
| ARIMA(0,1,1) | 1109886 | -0.30 | 1496325 | 4.55 | 3755137 | -7.23 | 4093961 | 7.23 |
| XGBoost | 2817876 | -5.91 | 3209634 | 10.39 | 2779742 | -5.39 | 3195985 | 5.38 |

ARIMA: Autoregressive Integrated Moving Average

MAE: Mean Absolute Error

MPE: Mean Percentage Error

MAPE: Mean Absolute Percentage Error

RMSE: Root Mean Square Error

XGBoost: eXtreme Gradient Boosting.

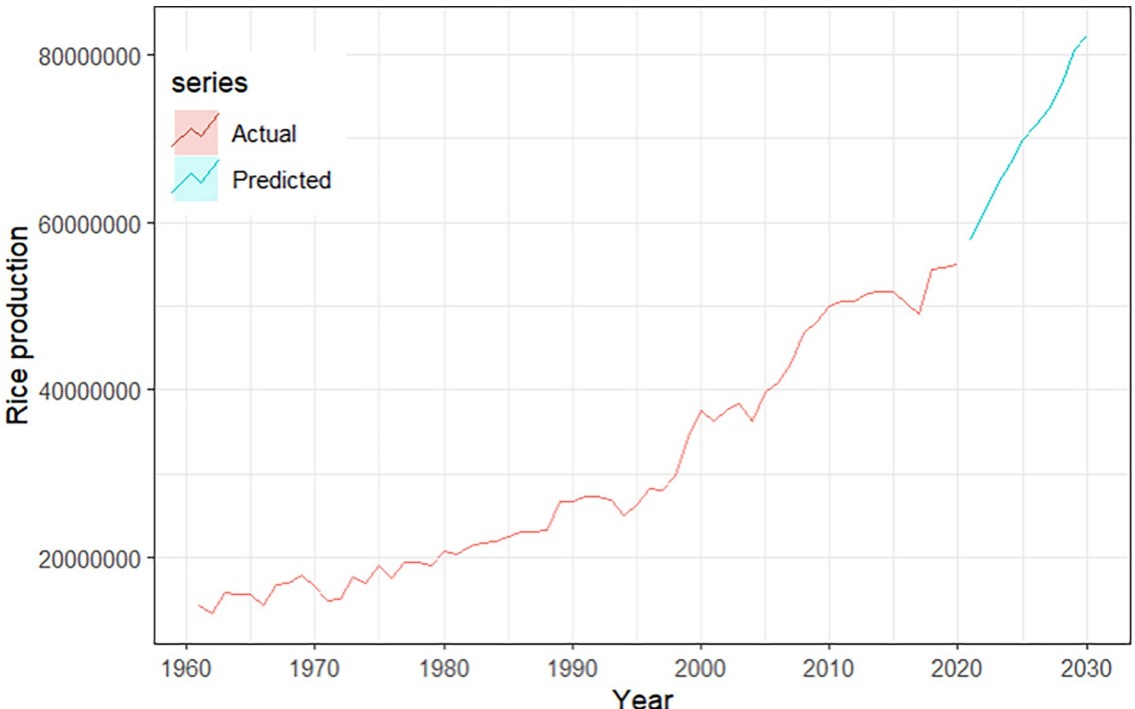

**Fig 10. Ten years' prediction of annual rice production in Bangladesh using XGBoost model.** XGBoost: eXtreme Gradient Boosting.

34°C annually, with an average temperature of roughly 26°C [48, 49]. Food production (e.g. rice, wheat) is particularly vulnerable to climate change because the agricultural productions are severely impacted by the climate patterns. Several previous studies examined that mean temperature can negatively impact the rice production [50, 51]. Precipitation had a positive impact on rice production, which was also determined by a previous study [52]. To know the actual pattern of the annual rice production in Bangladesh and forecast it accurately, time series modeling is very crucial [53].

The ARIMA model for the annual rice production data was established based on the concept of linear regression to forecast future data points. Without using any other explanatory variable, the ARIMA model is capable of understanding the pattern of the historical data and making accurate forecasts. So, it is simple to establish the ARIMA model [24]. Since ARIMA is a well-known and most widely used time series forecasting model, this study compared the ARIMA model with the robust XGBoost machine learning model. The ARIMA model can be well fitted to non-stationary data after the Box-Cox transformation and differencing of the original data [39]. But differencing can cause data lose. In order to differencing the data, this study lost one-year data. We built the ARIMA models using the auto.arima function by adjusting the power transformation parameter (lambda) and selected the appropriate model based on the lowest AICc value. Based on the lowest AICc value, we finally selected the optimal ARIMA (0,1,1) with the drift model.

On the other hand, we used the tree-based ensemble XGBoost supervised machine learning technique on our data. Several previous studies used several machine learning models, such as the artificial neural network [22], the random forest [26, 54, 55], and the support vector machine [56, 57] to predict rice production and obtained effective predicting results. The eXtreme gradient boosting is a robust machine learning technique for precisely modeling,

analyzing, and forecasting time series data [25]. The XGBoost model provides a variety of advantages regarding model forecasting. For example, it does not require any preprocessing of the data. It has a rapid processing speed, robust feature selection, good fitting, greater predictive performance and late scaling penalty than a typical Gradient boosting decision tree which removes the model from the occurrences of overfitting [25, 58]. As a result, we compared the predictive performance of the ARIMA model with the XGBoost model. From the result, it is clear that XGBoost performs better than the ARIMA model. In the meantime, the XGBoost model may also be utilized for cross-validation and has the ability to automatically identify significant feature vectors. The MAPE value of the XGBoost model for the test set is comparatively lower than the ARIMA model, which indicates XGBoost performs better than the ARIMA model in predicting the annual rice production in Bangladesh. Therefore, we used the XGBoost model to make a short-term prediction for the next 10 years. The prediction reveals that the amount of rice produced annually will grow in the following years in Bangladesh.

According to our study, the fitting and forecasting accuracy of the XGBoost model is much better than the traditional time-series ARIMA model. Without requiring any influencing factor, our proposed model can feasibly predict the annual rice production in Bangladesh.

## Limitations

In this study, we identified a model by comparing the ARIMA and XGBoost models that could accurately predict the annual rice production in Bangladesh. There are several machine learning models such as Decision Tree, LightGBM, and so on that are more robust and might have greater prediction accuracy. These models need to be applied in the future to find the best one. We mainly concentrated on the effect of time on rice production, which made it simpler to develop and predict our model. As a result, one of the limitations is that some climatic and econometric factors like temperature, rainfall, consumption, and so on, which are well known to affect rice production, were not taken into account in this study. These should be investigated further in light of the data's availability.

## Conclusion

We built an ARIMA and XGBoost model for forecasting the annual rice production in Bangladesh. These models were applied to generate a short-term prediction in this study. The XGBoost model performed better than the ARIMA model in predicting the annual rice production in Bangladesh. Finally, the government and development practitioners can employ XGBoost models over ARIMA to make more accurate short-term predictions of future crop production.

## Supporting information

**S1 File.**
(DOCX)

## Acknowledgments

We are very much grateful to the reviewers for providing valuable instructions and suggestions to make the study more appealing.

## Author Contributions

**Conceptualization:** Mst Noorunnahar.

**Data curation:** Mst Noorunnahar, Arman Hossain Chowdhury, Farhana Arefeen Mila.

**Formal analysis:** Mst Noorunnahar, Arman Hossain Chowdhury.

**Investigation:** Mst Noorunnahar, Farhana Arefeen Mila.

**Methodology:** Mst Noorunnahar, Arman Hossain Chowdhury.

**Resources:** Mst Noorunnahar, Arman Hossain Chowdhury, Farhana Arefeen Mila.

**Software:** Arman Hossain Chowdhury.

**Supervision:** Mst Noorunnahar, Farhana Arefeen Mila.

**Validation:** Mst Noorunnahar, Arman Hossain Chowdhury, Farhana Arefeen Mila.

**Visualization:** Mst Noorunnahar, Arman Hossain Chowdhury, Farhana Arefeen Mila.

**Writing – original draft:** Mst Noorunnahar, Arman Hossain Chowdhury, Farhana Arefeen Mila.

**Writing – review & editing:** Mst Noorunnahar, Arman Hossain Chowdhury, Farhana Arefeen Mila.

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
