## [Decision Letter · Decision Letter 0]

12 Dec 2022

PONE-D-22-20989Accuracy Performance of Time Series and Machine Learning Models for Predicting Rice Production in Bangladesh: A Comparative AnalysisPLOS ONE

Dear Dr. Mila,

Thank you for submitting your manuscript to PLOS ONE. After careful consideration, we feel that it has merit but does not fully meet PLOS ONE’s publication criteria as it currently stands. Therefore, we invite you to submit a revised version of the manuscript that addresses the points raised during the review process. Title of the manuscript should be changed highlighting the core idea of the study. Results and comparitive analysis should be improved.

We look forward to receiving your revised manuscript.

Kind regards,

Sathishkumar V E

Academic Editor

PLOS ONE

Journal Requirements:

Alim M, Ye GH, Guan P, Huang DS, Zhou BS, Wu W. Comparison of ARIMA model and XGBoost model for prediction of human brucellosis in mainland China: a time-series study. BMJ Open. 2020 Dec 7;10(12):e039676. doi: 10.1136/bmjopen-2020-039676. PMID: 33293308; PMCID: PMC7722837.

Rahman MS, Chowdhury AH, Amrin M (2022) Accuracy comparison of ARIMA and XGBoost forecasting models in predicting the incidence of COVID-19 in Bangladesh. PLOS Glob Public Health 2(5): e0000495. https://doi.org/10.1371/journal.pgph.0000495

In your revision ensure you cite all your sources (including your own works), and quote or rephrase any duplicated text outside the methods section. Further consideration is dependent on these concerns being addressed.

Reviewers' comments:

Reviewer's Responses to Questions

**Comments to the Author**

1. Is the manuscript technically sound, and do the data support the conclusions?

Reviewer #1: Partly

Reviewer #2: Partly

2. Has the statistical analysis been performed appropriately and rigorously? 

Reviewer #1: No

Reviewer #2: No

3. Have the authors made all data underlying the findings in their manuscript fully available?

Reviewer #1: Yes

Reviewer #2: Yes

4. Is the manuscript presented in an intelligible fashion and written in standard English?

Reviewer #1: No

Reviewer #2: Yes

5. Review Comments to the Author

Reviewer #1: Line number is missing which made the review difficult. The language of the manuscript needs fine tuning.

“The common previously used methods such as random forest and neural network still cannot handle missing values.” Please provide supporting literature for this statement.

Rice yield prediction has been widely studied all over the world. Please improve the introduction section by discussing about the various studies done over the world on rice yield forecasting.

“The XGBoost model with the greatest result for time series data was developed by changing the parameters frequently.” What are the ranges of the parameters tried to get the best result? Please mention that.

Results

“We found the presence of heteroscedasticity and non-normality in the data.” Please present the results of statistical analysis done to test the heteroscedasticity and non-normality of the data. At the same time, present the analysis results after boxcox transformation.

Discussion

“we found an increasing linear trend for the annual rice production data from 1961 to 2020 in Bangladesh.” I suggest the authors to analyse the trend in the rice yield data using Mann-Kendall or linear trend analysis.

Delete “To train these models, we used 90% of our data as training set and test the performance of the model using the remaining 10% of the data.”

Discussion is merely summary of the study. You should compare your results with previously published literature.

Reviewer #2: The study “Accuracy Performance of Time Series and Machine Learning Models for Predicting Rice Production in Bangladesh: A Comparative Analysis” is interesting. The study is well organized and executed properly, however, attention should be given to the following highlighted points before resubmission.

1. The abstract is verbose and does not highlight the results. It should report results and main findings instead of being generic.

2. The authors may provide some more detailed information regarding the XGBoost model which will be helpful for readers.

3. The authors mentioned that they used auto.arima function for selecting the best ARIMA model. “The ARIMA models were built with the 'forecast' package using auto.arima function for choosing the best model based on the AICc values [34]”. while the authors also mentioned this statement as well. “We performed the ADF test to see the stationarity of the data and found the data non stationary (p>0.01). To compensate for the trend shift observed in (Fig 3), we used first-order differencing of the sequence (Fig 4). The differenced time series was found stationary using the ADF test (p<0.01). So, the parameter of the ARIMA model d was 1”. In the ACF diagram, there was an evident peak at lag 1 indicating that the MA may become 1 and an evident spike at lags 0 in the PACF diagram, suggesting that the AR may become 0 (Fig 5). The authors may clearly state which procedure they used to choose the best ARIMA model.

4. What are the reasons that the authors may choose the ARIMA model with drift? This means defining the characteristics of the data.

5. Replace Table 1 and provide the P-values of the parameters and also the complete statistics. Secondly, the authors may also update the information for XGBoost model. The tuning parameters, etc.

6. “We used 8 time-lagged variables as input features for XGBoost; hence, the remaining 46 values were compared for the XGBoost model”. How the 8-time lagged is selected for XGBoost.

7. In Table 2 I feel some doubt about reporting the results. For the testing set, the results are consistent for all accuracy criteria. While for the training set the MAE value for XGBoost is very high. In the majority of cases, the MAE value is less than the RMSE value. Please check and rectify.

8. In the goodness of fit criteria, the authors may also include the directional statistics (DS) and Diebold Marino test (DM). Secondly, the MAPE results can be explained within its theoretical bounds. The authors may take help from the following studies. https://doi.org/10.1155/2020/1325071 and 10.1109/ACCESS.2019.2946992

6. PLOS authors have the option to publish the peer review history of their article (what does this mean?). If published, this will include your full peer review and any attached files.

Reviewer #1: **Yes: **Bappa Das

Reviewer #2: No

---

## [Author Response · Author response to Decision Letter 0]

19 Jan 2023

Dear Reviewers,

Greetings of the day. We are appreciative to the reviewers and editors for their insightful advice on how to improve our paper. We have meticulously reworked each portion of the article based on the reviewers' and editors' feedback. According to the authors' decision, we have also altered the title of the manuscript. We have substantially revised the entire manuscript. We believe that the modifications made to the new version will be acceptable.

Please let me know If you need any other necessary documents or corrections.

I am looking forward to hearing from you soon.

Thank you once again.

Best Regards

Farhana Arefeen MIla

---

## [Editor Report · Decision Letter 1]

8 Mar 2023

A tree based eXtreme Gradient Boosting (XGBoost) machine learning model to forecast the Annual Rice Production in Bangladesh

PONE-D-22-20989R1

Dear Dr. Mila,

We’re pleased to inform you that your manuscript has been judged scientifically suitable for publication and will be formally accepted for publication once it meets all outstanding technical requirements.

Kind regards,

Sathishkumar V E

Academic Editor

PLOS ONE
---

## [Editor Report · Acceptance letter]

14 Mar 2023

PONE-D-22-20989R1 

A tree based eXtreme Gradient Boosting (XGBoost) machine learning model to forecast the Annual Rice Production in Bangladesh 

Dear Dr. Mila:

I'm pleased to inform you that your manuscript has been deemed suitable for publication in PLOS ONE. Congratulations! Your manuscript is now with our production department. 

Kind regards, 

on behalf of

Dr. Sathishkumar V E 

Academic Editor

PLOS ONE